# An Edge Detection Method Based on Local Gradient Estimation: Application to High-Temperature Metallic Droplet Images

Ranya Al Darwich *, Laurent Babout * and Krzysztof Strzecha *

Institute of Applied Computer Science, Łódź University of Technology, 90-537 Łódź, Poland

* Correspondence: raldarwich@kis.p.lodz.pl (R.A.D.); laurent.babout@p.lodz.pl (L.B.); strzecha@kis.p.lodz.pl (K.S.)

**Abstract:** Edge detection is a fundamental step in many computer vision systems, particularly in image segmentation and feature detection. There are a lot of algorithms for detecting edges of objects in images. This paper proposes a method based on local gradient estimation to detect metallic droplet image edges and compare the results to a contour line obtained from the active contour model of the same images, and to results from crowdsourcing to identify droplet edges at specific points. The studied images were taken at high temperatures, which makes the segmentation process particularly difficult. The comparison between the three methods shows that the proposed method is more accurate than the active contour method, especially at the point of contact between the droplet and the base. It is also shown that the reliability of the data from the crowdsourcing is as good as the edge points obtained from the local gradient estimation method.

**Keywords:** edge detection; active contour; gradient-based edge detection; crowdsourcing





## 1. Introduction

Edges have been defined as significant local variations in image intensity and are considered an essential feature of image analysis. Given the importance of edge detection in image processing, image analysis, image pattern recognition and computer vision techniques, edge detection has continued to be an effective research area and a fundamental step in retrieving information from digital images [1]. There are many edge detection methods that have been developed by several researchers [2–4]. However, there are some factors that affect the performance of the edge detection process, especially in the case of images with low-intensity differences between adjacent regions. These differences lead to blurring of the boundaries between these regions. Blurred boundaries can affect the accuracy of the edges delineation and make edge detection a challenging task [5]. This paper addresses the problem of detecting edges in images of hot specimens of molten copper that emit thermal radiation since the illumination of the specimen's boundaries can be considered an additional factor affecting image quality [6].

The aim of the research in this paper is to propose an edge detection method based on the local gradient estimation to define the edges of a metallic droplet images obtained from the THERMO-WET system for high temperature measurements of surface properties and compare the edges to the contour line obtained from the active contour model of the same images [7], and crowdsourcing-based detection [8].

The studied images show that the edges are noticeably blurred due to the thermal effects. The edge detection method presented in this paper prove to give better results compared to the classic active contour method [9].

## 2. Materials and Methods

The studied images were obtained from the THERMO-WET system, as the first in the world, which enabled automated measurement in a wide temperature range (up to

1800 °C) in a protective atmosphere [10]. The developed system is able to measure the surface tension of the liquid phase, and the extreme wetting angle of the base by the liquid.

Several methods have been developed to measure interfacial tensions and contact angles, such as the well-known axisymmetric drop shape analysis (ADSA) method [11], electrowetting-on-dielectric (EWOD) [12]. These methods can only be used for symmetrical droplets. However, in most cases, the specimen is not symmetric. Therefore, there was a need to develop methods that could measure non-axisymmetric droplets. These methods include polynomial fitting [13], sub-pixel polynomial fitting (SPPF) [14], moving goniometric mask [15], and 3D analysis [16].

The measurement method related to THERMO-WET system is based on image processing and analysis algorithms, and is conducted using the sessile drop technique [17]. In short, the method works as follows. The camera records and monitors images of the melting process of the specimen (in this study, this specimen is copper) placed inside the high-temperature furnace. The specimen begins to melt at 1080 °C, and takes a spherical shape at about 1134 °C. The measurements are performed in a temperature range of 1135–1300 °C with a 20 °C step. Twenty images are acquired for each temperature. There are two forces that affect the shape of a droplet placed on a non-wettable surface: the surface tension, which attempts to give the droplet a spherical shape, and gravity by which the droplet is flattened [18].

The specificity of images taken at high temperatures makes the segmentation process particularly difficult and requires the use of specialized, dedicated algorithms. Figure 1 shows an original image of a copper specimen obtained from the THERMO-WET measurement system at a high temperature (1175 °C). The image shows low contrast and blurry boundaries, which makes image segmentation a challenging task.

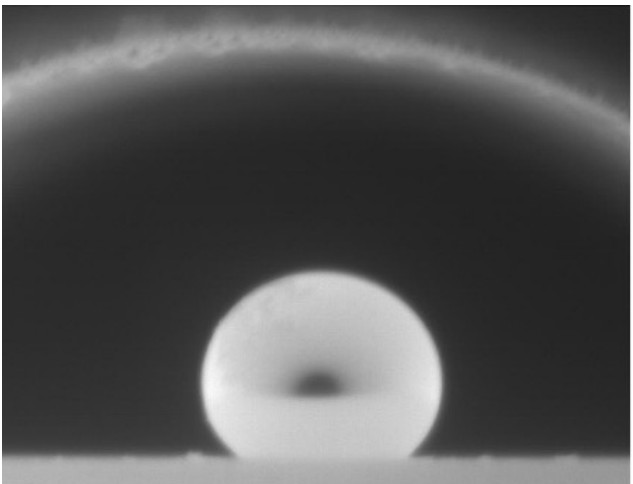

**Figure 1.** Image of a copper specimen obtained from THERMO-WET system at (1175 °C).

### 2.1. Gradient-Based Edge Detection

Edge detection is basically the process of detecting significant local variations in an image. The concept of image gradient is the first and most intuitive approach for extracting image edge pixels, as it is a measure of the contrast in the intensity of an image. It is defined as a two-dimensional vector equivalent to the first derivative, where the magnitude of this vector indicates the edge strength and its angle indicates the direction of the edge [19].

Gradient-based edge detection method aims to detect edges by finding the maximum and minimum in the first derivative of an image, where pixels with a high gradient are considered edges, i.e., image pixels whose intensity are locally maximum in the direction of the gradient are labeled as edges [20]. In this paper, we propose a gradient-based edge detection method to detect the edges by finding the maximum and minimum in the first derivative of an image.

### 2.2. Active Contour Model

The active contour model, or snakes, is one of the most effective tools in image segmentation, proposed by Kass and Witkin [21]. Since their introduction, active contour models have undergone extensive research to detect boundaries. Caselles, Kimmel, and Sapiro [22] have proposed new improved versions of the python model called the active geodetic engineering contour model (GAC). Active contour consists of developing contour in images toward the boundaries of objects. It relies on strong mathematical properties and strong numerical charts based on the placement method [23]. The basic snake pattern is a deformed elementary shape that is affected by the forces of external constraint and image forces that pull it toward the perimeter of the object, and internal forces that resist deformation. The image forces push the snake to the boundaries of the objects, while external constraint forces place the snake close to the required local minimum [24].

### 2.3. Crowdsourcing-Based Detection

Over the past decade, crowdsourcing has been widely used as a method to solve problems that require human intelligence [25]. In this paper, we present a crowdsourcing method to instruct people to locate the edges of metallic droplet images at specific points. Studies have shown that humans can effectively define the droplet edges compared to the results obtained from the local gradient estimation method and the active contour segmentation method. Participants were given the instructions to place characteristics points detailed later in the paper on a set of images without prior knowledge about the segmentation results. In that matter, the information retrieved from this manual repetitive approach represents the so-called ground truth of this study.

### 2.4. Proposed Algorithm Pipeline

First, images of the copper specimen were grouped into three temperature-based sets of five images of the same temperature. Each grey scale image was converted to a binary image using the Otsu thresholding method [26], which automatically searches for the optimum threshold intensity. It works by finding the threshold intensity that optimally separates the image into two classes: the foreground and the background. It does this by minimizing a metric of within-class variance [27]. Otsu's method was proposed for this case because the difference in the variances of the droplet and the background is significant; in another word, the droplet is brighter compared with the background [28]. The binary image generated by the Otsu method will be used as a mask for the active contour segmentation method. The mask specifies the initial status of the active contour. The reason for using the Otsu threshold method to create the mask was to obtain faster and more accurate segmentation results, by setting an approximate shape and starting position near the desired object boundaries. The method used is the Chan–Vese active contour model to detect objects without sharp edges, the Chan–Vese model based on techniques of curve evolution, Mumford–Shah functional for segmentation and level sets [29], with two parameters one of them "SmoothFactor = 0.01" that controls the smoothness of the boundaries of the segmentation region, and the other parameter "ContractionBias = 0.01" which affects the direction of the contour to grow outward or shrink inward as is illustrated in Figure 2. The software was implemented in a MATLAB environment.

Figure 3 shows the image after implementing the edge function, which returns a binary image, where the edges of the droplet are white and everything else is black. Edge function uses the Sobel edge detection method. The Sobel operator is a discriminant operator, used to find the approximation of a derivative of an image intensity value [30]. It is easy to implement, compared to other operators. It is based on convolving the image with a $3 \times 3$ filter in horizontal and vertical direction and is therefore relatively inexpensive in terms of calculations [31].

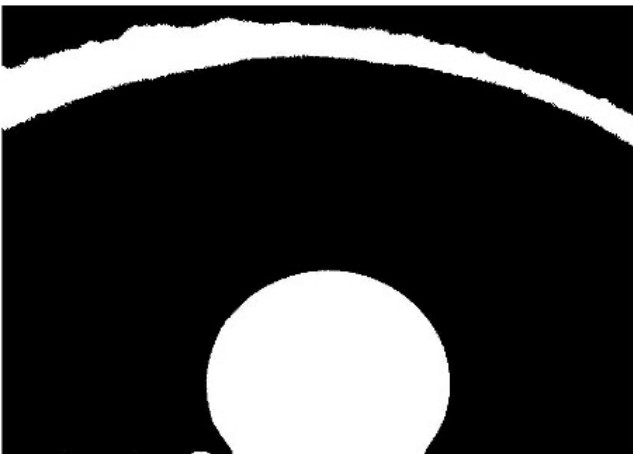

**Figure 2.** The segmented image using active contour model.

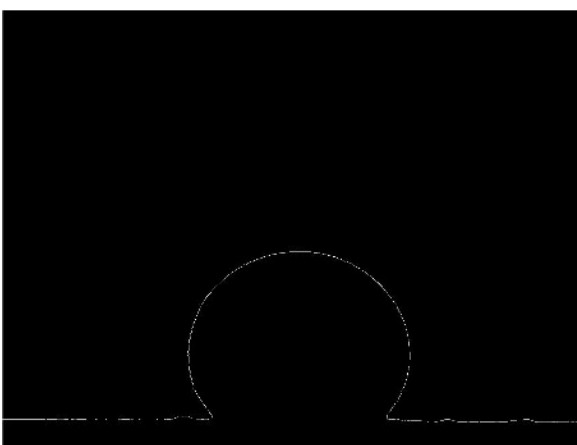

**Figure 3.** The segmented image after implementing Sobel's method of edge detection.

This boundary serves as initial data for the iterative method described in this paper. From this approximation edge line, the key points representing the top of the droplet (S) and the contact point between the droplet and the base on the left side (E) are automatically selected to find the corresponding midpoint (M). The gradient is calculated along the bisector passing through (M), where the pixels with the largest gradient values are the possible edge pixels as shown in Figure 4a. This detected point is used as either an (S) or (E) point in the next iteration, and the midpoint and edge detection process continues from the gradient profile as shown in Figure 4b.

By repeating the previous process between each two successive edge points starting at (S) and ending with (E), we obtain the edge points shown in Figure 4c,d. Figure 4e shows the same previous process on the right side of the droplet. Figure 4f shows the edge points obtained from the gradient method with the active contour line, while Figure 4g shows the retrieved edge points on the original image. Figure 4h shows the edge points along the boundaries of the droplet as well as the contact area between the droplet and the base.

To obtain the edge points in the contact area between the droplet and the base, the same process was used but with different reference points as shown in Figure 5, where Figure 5a shows the edge points obtained from the gradient method and the active contour line, and Figure 5b shows the edge points on the original image. We can see that the proposed edge detection method based on local gradient estimation performs better than the state-of-the art procedure.

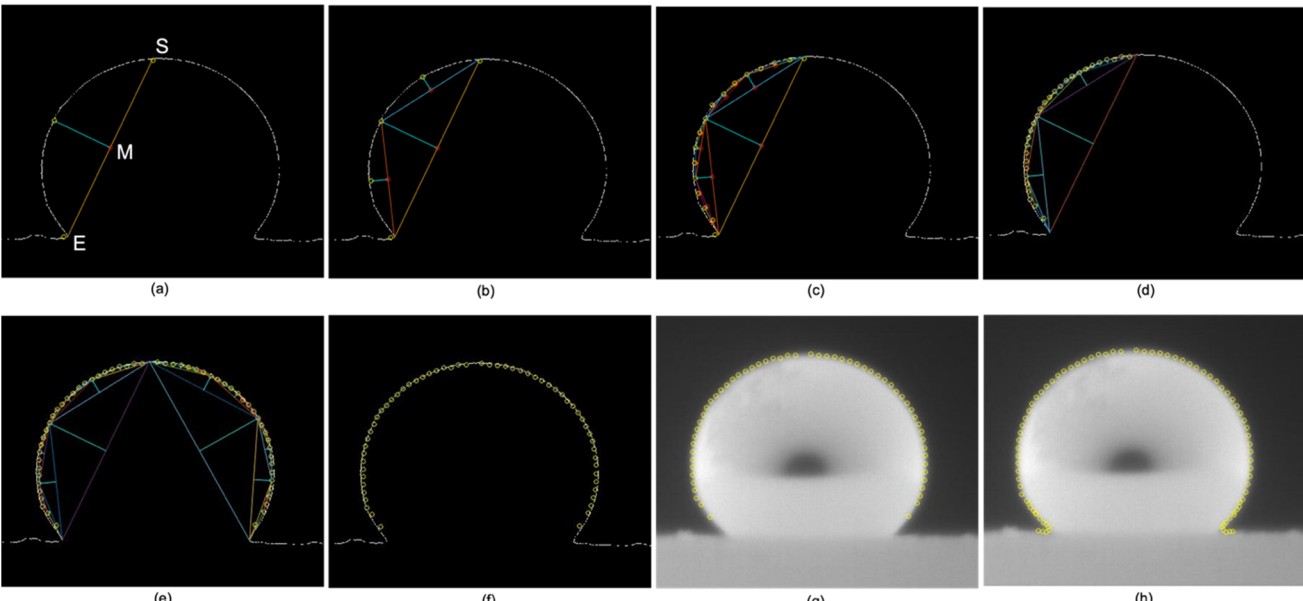

**Figure 4.** Steps to get droplet edge points using the local gradient estimation method. (**a**) shows the key points representing the top of the droplet (S) and the contact point between the droplet and the base on the left side (E), the corresponding midpoint (M), and the edge pixel (in yellow). (**b**–**d**) show a repetition of the previous process, with edge pixels as key points. (**e**) shows the same previous process on the right side of the droplet. (**f**) shows the retrieved edge points and the active contour line. (**g**) shows the retrieved edge points on the original image. (**h**) shows the edge points along the boundaries of the droplet as well as the contact area between the droplet and the base.

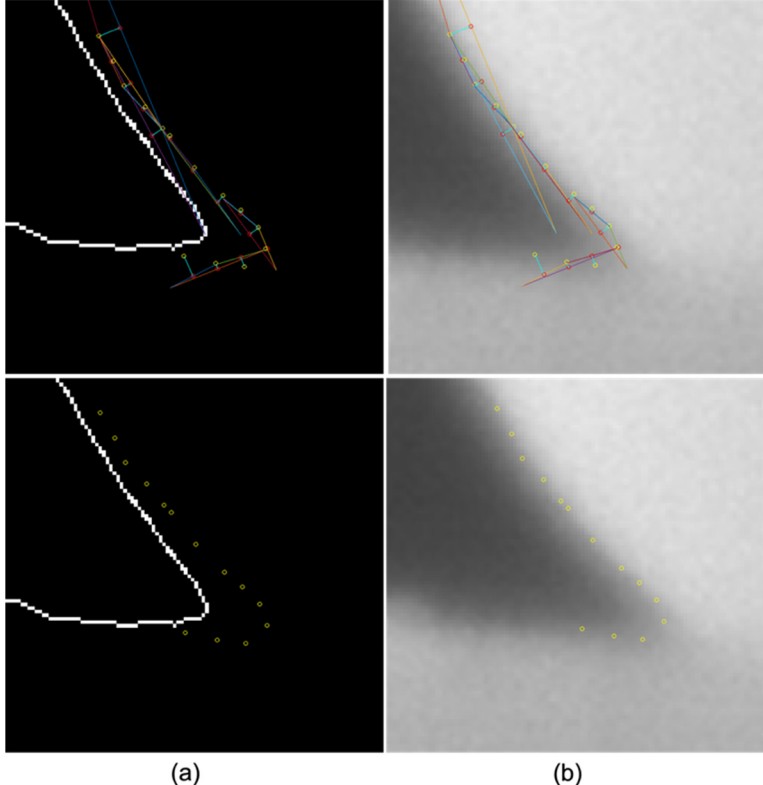

**Figure 5.** The edge points in the contact area between the droplet and the base on the left side of the droplet. (**a**) shows the edge points obtained from the gradient method and the active contour line. (**b**) shows the edge points on the original image.

The pseudo code for the edge detection algorithm based on local gradient estimation is given below in Algorithm 1.

---

**Algorithm 1:** An edge detection algorithm based on local gradient estimation

---

**Input:** *IMAGE*, an image of a copper specimen;
**Output:** *OUT*, a set of edge points;
    **Convert** a grayscale image to a binary image based on threshold;
**Call** activecontour with image, mask;
**Call** edge with segmented image;
**Find** *LIND* indices and values of nonzero elements on the edge line;
**Return** the matrices r and c containing the equivalent row and column in the matrix *LIND*;
**Find** *S* the point on top of the droplet;
**Find** *E* the contact point between the droplet and the base on the left side;
**Compute** *M* the midpoint between *S* and *E*;
**Calculate** *Grad* the gradient along the bisector passing through *M*;
*OUT* ← max(*Grad*);
**Repeat** the previous process between each two successive edge points starting at (S) and ending with (E),
**Return** *OUT*;

---

## 3. Results

The edge points were compared with the ground truth case based on the manual detection of key boundary points by 15 volunteers. All results are presented in Figure 6, which confirms that the proposed approach shows results similar to the crowdsourcing-based approach.

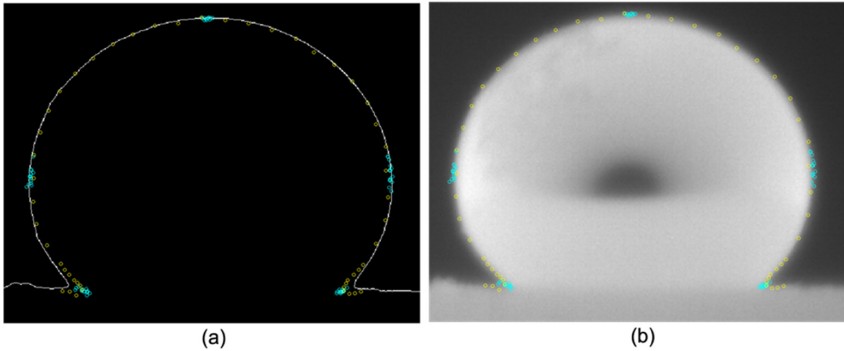

(a)          (b)

**Figure 6.** Comparison between the 3 methods. (**a**) Active contour (continuous line), ground truth (blue points), the proposed method (yellow points). (**b**) Superposition of points from the ground truth and the proposed method on the original image.

Table 1 summarizes the differences between the active contour method and the local gradient estimation method for a single image, for points of interest (highest point, left equator point, right equator point, left contact point, right contact point) as shown in Figure 7, and compare the two methods with the ground truth, which is the average value of the results obtained by volunteers. The comparison is made between the coordinates of the points of interest, i.e., the row and column of each point of interest for the three methods. The table shows that the coordinate values of the edge points obtained from the proposed method are very similar to the ground truth, unlike the edge point obtained from the active contour method, which proves that the proposed method is more accurate than the active contour method.

The results in Table 1 show that the method based on local gradient estimation method and ground truth (crowdsourcing method) is similar with slight differences in a few pixels and closer to the droplet boundary at points of interest, compared to the active contour method. These differences significantly increase in the contact area between the droplet and the base.

**Table 1.** The coordinate values of the three methods for the points of interest (highest point, left equator point, right equator point, left contact point, and right contact point).

| Methods | Active Contour | Local Gradient Estimation Edge Detection | Ground Truth (Mean Value) |
|---|---|---|---|
| Highest point | P1 [646, 535] | P1 [661, 538] | P1 [653, 537] |
| Left equator point | P2 [412, 757] | P2 [418, 755] | P2 [415, 752] |
| Right equator point | P3 [903, 757] | P3 [899, 749] | P3 [901, 749] |
| Left contact point | P4 [466, 904] | P4 [483, 908] | P4 [487, 911] |
| Right contact point | P5 [853, 905] | P5 [838, 908] | P5 [835, 911] |

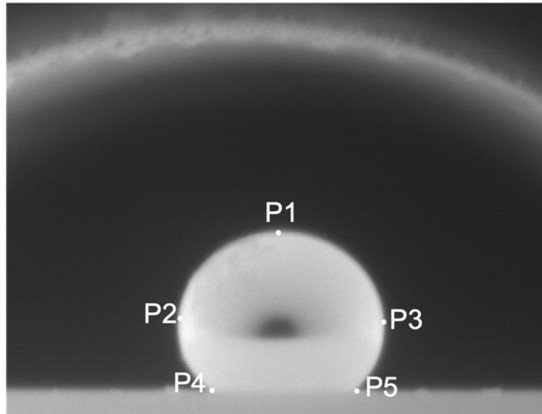

**Figure 7.** The points of interest: P1 (highest point), P2 (left equator point), P3 (right equator point), P4 (left contact point), and P5 (right contact point).

Figure 8 shows a comparison of three different images of the same specimen taken at different temperatures at 1175 °C, 1133 °C, and 1234 °C for images 1, 2, and 3, respectively. The images in the upper part show the edge points on the original images, and the images in the lower part show the edge points obtained from the gradient method and the active contour line.

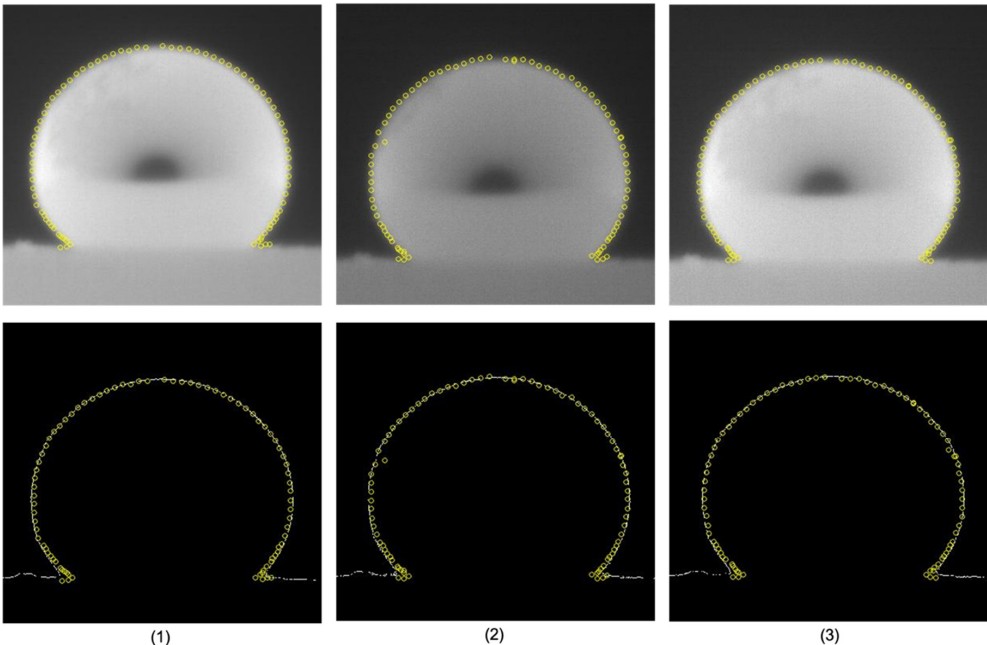

**Figure 8.** Comparison of three different images of the same specimen taken at different temperatures. (**1**) an image of the droplet taken at 1175 °C, (**2**) an image of the droplet taken at 1133 °C, (**3**) an image of the droplet taken at 1234 °C.

The results presented above indicate that the proposed method can be implemented on more than one image and gives accurate localization of the edge points of the droplet. The proposed approach could be developed in the future for a large range of images. Moreover, the extracted point set can be used to measure the contact angle based on the ellipse approximation method proposed in [18] and check whether it improves the calculated values significantly. This work is in progress.

## 4. Conclusions

In this paper, we proposed a local gradient estimation method for edge detection of metal droplet images captured at high temperatures. Since the considered images have low contrast and blurry borders, it was difficult to achieve segmentation success. However, the active contour model has been considered one of the most efficient segmentation methods. It should be noted that the local gradient estimation method proposed in this paper showed better results compared to the active contour model, where the edge points are closer to the droplet edges than the active contour curve. The experimental results show that the proposed method is accurate and effective, even with low contrast and blurry conditions, compared to the results obtained from crowdsourcing-based detection, and it may find wide use in related fields.

**Author Contributions:** Conceptualization, R.A.D. and L.B.; methodology, R.A.D. and L.B.; software, R.A.D.; validation, R.A.D.; investigation, R.A.D. and K.S.; writing—original draft preparation, R.A.D.; writing—review and editing, R.A.D., K.S. and L.B.; supervision L.B. All authors have read and agreed to the published version of the manuscript.

**Funding:** The work was financed from the statutory funds of the Institute of Applied Computer Science, Lodz University of Technology, Poland, No. 501/2-24-1-211.

**Conflicts of Interest:** The authors declare no conflict of interest.

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
