# Peer review of "An Edge Detection Method Based on Local Gradient Estimation: Application to High-Temperature Metallic Droplet Images"

_applsci, doi:10.3390/app12146976_

Round 1

Reviewer 1 Report

This manuscript proposes novel method for edge detection of images with low contrast.  I cannotgive green light for further processing of this manuscript without some heavy revisions, which in my opinion could not be done in a meaningful way and timely manner if I choose major revision. Therefore I must advise reject and encourage resubmission. The aspects which made me give such review are following:

  • Edge detection is such a broad field. From gradient based method to energy based methods to distribution based methods, I think the authors should first thoroughly re-read many of the articles available to get more insight if their method gives any novelty to the field.
  • There is enormous misunderstanding between segmentation and edge detection.  The image segmented as shown in Figure 2 will not most certainly results in edges as shown in Figure 3. This is not how it works. Maybe the captions are wrong. If the Figure 3 gives the sobel edge detection it should be given more information of other parameteres needed. If the otsu method is used to find threshold it should be noted how the otsu method finds a threshold? On grayscale values of gradient estimations?
  • The edge detection method presented here, assumes the images are almost all the same, meaning that droplet must occur, the droplet must be pointing upwards etc... They are referencing something named "THERMO-WET", which I am not familiar, however after observing the reference [10], which was given for THERMO-WET system, I could not find any reference on how the wetting works or how the measurments are made to ensure that the image would be practically the same as the one given here.  I think if the presented novel edge detection method solves this problem a little more attention should be given to this topic. Also, after observing the papers that also cite the Ref. [10], I found the reference Computer vision system for high temperature measurements of surface properties, by same authors as Ref. [10] which provide very similiar method. It occurs to me that you should at least mention such methods which are also specialized in image processing of THERMO-WET images.
  • The comparisons provided here are lacking. The authors give comparions with only one method, which they also use as a part of their method, hence, the improvements are expected, otherwise the method would be illogical. Finally are we comparing the effectivness of edge detection or how good this system works for measuring. This is distinctive, as the we compare edge detection methods using standards metrics such as precision, accuracy, Dice, etc. Here, the authors reported the positions of isolated points which may have importance for measurments, however, I find them losely related to the edge detection success.

Author Response

We would like to thank the reviewer for his thoughtful comments and efforts towards improving our manuscript. In what follows, we highlight the reviewer comments and questions, and our response to each comment and question.

Comment1: “The image segmented as shown in Figure 2 will not most certainly results in edges as shown in Figure 3” 

Response 1: Caption in Figure 3 was wrong and has been corrected. In line 140 of the manuscript.

Comment2: “If the Figure 3 gives the sobel edge detection it should be given more information of other parameters needed.”

Response 2: Detailed information about Sobel operator is given in lines 131-136 in the manuscript.

Comment3: “If the Otsu method is used to find threshold it should be noted how the Otsu method finds a threshold? On grayscale values of gradient estimations?”

Response 3: Otsu's thresholding method automatically searches for the optimal threshold intensity. It works by finding a threshold intensity that optimally separates an image into foreground and background. It does this by minimizing a metric of within-class variance. This question is addressed in lines 109–119 of the manuscript.

Comment 4: “The edge detection method presented here, assumes the images are almost all the same, meaning that droplet must occur, the droplet must be pointing upwards etc... “

Response 4: The studied images are not the same, there is a slight difference between them. However, we do agree that the point selection (top, equator, base) assumes a specific shape. Still, it should to be clear that these points are selected automatically.

Comment 5: “They are referencing something named "THERMO-WET", which I am not familiar, however after observing the reference [10], which was given for THERMO-WET system, I could not find any reference on how the wetting works or how the measurements are made to ensure that the image would be practically the same as the one given here.  I think if the presented novel edge detection method solves this problem a little more attention should be given to this topic.”

Response 5: In fact, the reviewer is right. However, we explained how the wetting works in lines 55-62 of the manuscript.

Comment 6: “after observing the papers that also cite the Ref. [10], I found the reference Computer vision system for high temperature measurements of surface properties, by same authors as Ref. [10] which provide very similar method. It occurs to me that you should at least mention such methods which are also specialized in image processing of THERMO-WET images.”

Response 5: Thank you for your note. Several methods have been developed to measure interfacial tensions and contact angles, such as axisymmetric drop shape analysis (ADSA) method, electrowetting-on-dielectric (EWOD). Also, methods that can measure non-axisymmetric droplets. Including polynomial fitting, sub-pixel polynomial fitting (SPPF), moving goniometric mask, and 3D analysis can be found in the literature. We mentioned these methods in lines 45-53 of the manuscript.

Comment6: “The comparisons provided here are lacking. The authors give comparisons with only one method, which they also use as a part of their method, hence, the improvements are expected, otherwise the method would be illogical. “

Response 6: It is true that we have compared our method with the active contour method, but we use only two key points of the active contour line the as initial data for our method.

Comment 7: “Finally are we comparing the effectiveness of edge detection or how good this system works for measuring. This is distinctive, as the we compare edge detection methods using standards metrics such as precision, accuracy, Dice, etc. Here, the authors reported the positions of isolated points which may have importance for measurements.”

Response 7: We compared the effectiveness of edge detection between the active contour line and the edge points obtained from the local gradient estimation method and the ground truth presented by the crowdsourcing-based method. The proposed method is accurate and efficient and showed better results compared to the active contour model, where the edge points are closer to the edges of the droplet than the active contour curve.

Reviewer 2 Report

In this manuscript, the authors proposed a local gradient estimation method for edge detection of metal droplet images captured at high temperatures. The local gradient estimation method proposed in this manuscript shows better results compared to the active contour model, where the edge points are closer to the droplet edges than the active contour curve. The experimental results show that the proposed method is accurate and effective even with low contrast and blurry conditions compared to the results obtained from crowdsourcing-based detection.

The reviewer has the following comments and suggestions:

1. Section 2. “Materials and Methods” is more like an introduction. The authors should include the details of the three methods used in the manuscript, e.g., algorithm.   

2. The authors should define a parameter to describe the differences between the three methods for the points of interest. It will help the readers clearly know which method has the better performance.

3. The authors should discuss more about Fig. 8. For example, what information can we get from Fig. 8?

4. There are many mistakes and typos in the manuscript. For example,Figure 3 shows the image after implementing the edge function which returns a binary image containing ones where the function finds edges in the binary image and zeros elsewhere. edge uses Sobel's method of edge detection by default.”. And the English have to be improved.

Author Response

We would like to thank the reviewer for his thoughtful comments and efforts towards improving our manuscript. In what follows, we highlight the reviewer comments and suggestions, and our response to each comment.

Comment 1: “Section 2. “Materials and Methods” is more like an introduction. The authors should include the details of the three methods used in the manuscript, e.g., algorithm.”  

Response 1: Details of the three methods have been included in the “Materials and Methods” section.

Comment 2: “The authors should define a parameter to describe the differences between the three methods for the points of interest. It will help the readers clearly know which method has the better performance.”

Response 2: A parameter describing the differences between the three methods has been defined which is the coordinates of the points of interest. In lines 184-186 of the manuscript.

Comment 3: “The authors should discuss more about Fig. 8. For example, what information can we get from Fig. 8?”

Response 3: In fact, the reviewer is right. we added a discussion about Figure 8 in lines 211-216 of the manuscript.

Comment 4: “There are many mistakes and typos in the manuscript. For example, “Figure 3 shows the image after implementing the edge function which returns a binary image containing ones where the function finds edges in the binary image and zeros elsewhere. edge uses Sobel's method of edge detection by default.”. And the English have to be improved.”

Response 4: We totally agree. The mistakes and typos have been corrected.

In addition to the suggestion above, we are working to improve academic English.

Round 2

Reviewer 1 Report

The authors addressed  many comments appropriately. I can now suggest publishing of the article. I would however suggest improving image quality.

Reviewer 2 Report

Since all my concerns are addressed, I think this manucript can be accepted.